# Human Placental Adaptive Changes in Response to Maternal Obesity: Sex Specificities

**DOI:** 10.3390/ijms24119770

**Published:** 2023-06-05

**Authors:** Esther Dos Santos, Marta Hita Hernández, Valérie Sérazin, François Vialard, Marie-Noëlle Dieudonné

**Affiliations:** 1UFR des Sciences de la Santé Simone Veil, Université de Versailles-Saint Quentin en Yvelines—Université Paris Saclay (UVSQ), INRAE, BREED, F-78350 Jouy-en-Josas, France; marta.hita-hernandez@uvsq.fr (M.H.H.); valerie.serazin@ght-yvelinesnord.fr (V.S.); francois.vialard@uvsq.fr (F.V.); marie-noelle.dieudonne@uvsq.fr (M.-N.D.); 2Ecole Nationale Vétérinaire d’Alfort (EnvA), BREED, F-94700 Maisons-Alfort, France; 3Service de Biologie Médicale, Centre Hospitalier de Poissy-Saint Germain, F-78300 Poissy, France

**Keywords:** maternal obesity, placenta, environmental adaptation, sex specificities, reproduction

## Abstract

Maternal obesity is increasingly prevalent and is associated with elevated morbidity and mortality rates in both mothers and children. At the interface between the mother and the fetus, the placenta mediates the impact of the maternal environment on fetal development. Most of the literature presents data on the effects of maternal obesity on placental functions and does not exclude potentially confounding factors such as metabolic diseases (e.g., gestational diabetes). In this context, the focus of this review mainly lies on the impact of maternal obesity (in the absence of gestational diabetes) on (i) endocrine function, (ii) morphological characteristics, (iii) nutrient exchanges and metabolism, (iv) inflammatory/immune status, (v) oxidative stress, and (vi) transcriptome. Moreover, some of those placental changes in response to maternal obesity could be supported by fetal sex. A better understanding of sex-specific placental responses to maternal obesity seems to be crucial for improving pregnancy outcomes and the health of mothers and children.

## 1. Introduction

Obesity (excess adipose tissue) is characterized by a pro-inflammatory environment, profound dyslipidemia, and lipotoxicity in various organs. It constitutes one of the greatest public health challenges of the 21st century [1]. A worldwide survey found that the prevalence of obesity (defined as a body mass index [BMI] ≥30 kg/m^2^) has doubled since 1980 in more than 70 countries and is steadily increasing in most other countries [2]. Unsurprisingly, the prevalence of obesity during pregnancy is also rising.

Maternal obesity increases the risk of adverse health outcomes in both mothers and children. Complications of pregnancy include preeclampsia, gestational diabetes (GD) mellitus, and gestational hypertension [3]. Concerning the impact on children, it was shown that maternal obesity in the absence of GD did not affect birthweight but that it was specifically associated with greater adiposity at birth in female offspring (but not in male offspring) [4]. Moreover, with reference to the developmental origins of health and disease (DOHaD), there is evidence to suggest that maternal obesity increases the risk of the offspring suffering from metabolic and cardiovascular diseases (including obesity, type 2 diabetes mellitus, and metabolic syndrome) later in life [5,6,7,8]. Recent research indicates that male offspring are more susceptible to neurodevelopmental disorders in the case of maternal obesity [9].

The relationship between the mother’s health and the child’s health is closely linked to the placenta, which constitutes the primary site for maternal–fetal exchanges. A large body of data in the literature has highlighted the impact of an obesogenic maternal environment on the placenta’s development and exchange functions, although some of the results are subject to debate [10,11,12,13,14,15,16]. The heterogeneity of the study population in terms of lifestyle factors, dietary habits, ethnicity, and the prevalence of pregnancy-related complications (such as GD) could explain these discrepancies. Another possible explanation is that most studies did not consider fetal sex. Indeed, the idea that the sex of the fetus could influence the way the placenta adapts to its environment is still fairly recent [17,18,19].

In this context, and in order to reduce the influence of possible confounding factors, our review focuses on recent studies of the impact of maternal obesity (in the absence of GD) on the placenta’s main characteristics, such as its morphology, endocrine function, metabolism, and inflammatory/oxidative status. We also highlight the influence of fetal sex during pregnancy and describe some sex-specific placental adaptations to maternal obesity.

## 2. Placental Development

The placenta is a complex organ with an essential role in embryo implantation and the maintenance of pregnancy. More specifically, the human placenta is “hemochorial” and characterized by a direct contact of the placental cells with the maternal blood [20]. The placenta exerts many essential functions for fetal survival: (i) the formation of a physical and immune barrier between the maternal and fetal circulations, in order to protect the fetus from certain pathogens; (ii) the production of a multitude of signals (such as hormones and growth factors) that are crucial for maternal and fetal metabolism, and (iii) the exchange of nutrients, gases, and water between the mother and the fetus [21].

Placental cells can differentiate along two distinct pathways: the villous and extravillous pathways. In the villous pathway, mononuclear cytotrophoblasts (CTs) differentiate by fusing to form the syncytiotrophoblast (ST)—a specialized, multinuclear syncytium on the outer layer of placental villi. The ST will ensure fetal–maternal exchanges, hormone production, and protection against a number of pathogens [22]. In the extravillous pathway, CTs provide the placenta with anchorage in the maternal uterus. Indeed, CTs are able to proliferate and differentiate, in order to gain an invasive phenotype and penetrate into the maternal decidua and myometrium. The matrix metalloproteinases (MMPs) 2 and 9 enable the degradation of the decidual and endothelial extracellular matrix and, thus, the invasion by CTs [23]. Moreover, invasive CTs colonize the maternal spiral artery; this favors fetal development by expanding the vessel diameter, reducing vessel contractility, and enabling constant oxygen delivery at a low blood pressure [24]. The interactions between CTs and a range of uterine cell types (such as uterine natural killer (uNK) cells, lymphocytes, macrophages, dendritic cells, and decidual stromal cells) during the invasion have an important role in the immune acceptance of the placental/fetal allograft and in the timing and depth of CT invasion. The particular pattern of histocompatibility antigens expressed by invasive CTs appears to be one of the most important strategies; in contrast to most somatic cells, polymorphic human leukocyte antigen (HLA) class Ia molecules are not present at the surface of invasive CTs [25]. Indeed, invasive CTs express the non-classical major histocompatibility complex (MHC) class IB antigens, including HLA-G, with strong immunosuppressive effects. During placentation, invasive CTs are exposed to maternal immunocompetent cells in the decidual environment. This constitutes a real challenge in terms of immune modulation which is controlled, in part, by HLA-G. Indeed, soluble HLA-G is able to (i) promote the apoptosis of activated maternal CD8+ T cells, (ii) inhibit the proliferation of maternal CD4+ T cells, and (iii) abrogate uNK-cell-mediated cytotoxicity [26]. Some complications of pregnancy (such as spontaneous abortion and pre-eclampsia) are associated with the failure of CT invasion; this might be due to a premature rise in oxygen levels, which increases oxidative stress and thus damages the placental villi. In summary, correct placental development is the key to the successful progress of pregnancy.

## 3. Impact of Maternal Obesity on Placental Development: Trophoblast Differentiation

The chorionic villus is the human placenta’s structural and functional unit and it is covered by the ST, i.e., a highly specialized, multinucleated, epithelial cell layer [20]. Indeed, the ST is derived from CT fusion through a process called syncytialization. This process involves a membrane protein of retroviral origin (syncytin-2), which binds to its specific receptor (major facilitator superfamily domain containing 2, MFSD2) and enables CT fusion. Both syncytin-2 and its receptors are strongly expressed in the ST [27,28]. Furthermore, the CT’s differentiation into a functional ST is associated with the elevated production of hormones such as leptin, progesterone, human chorionic gonadotropin (hCG), and human placental lactogen (hPL) [20]. Thus, the ST underpins the placenta’s endocrine functions throughout pregnancy [29].

The results of two recent studies clearly showed that maternal obesity influences the placenta’s endocrine function. Firstly, with regard to the biochemical differentiation of placental cells, researchers have found that the levels of secretion of three key hormones (hCG, leptin, and progesterone) by the ST were abnormally low in placentas from obese women [10,30]. Since hCG and leptin are actively involved in the growth and development of the fetal–placental unit [31,32], one can hypothesize that a low production of these hormones disturbs fetal growth. Moreover, another adipokine (adiponectin, which is also secreted at the fetal–maternal interface by the endometrium itself) appears to favor the development of a functional placenta with differentiative abilities [33,34]. Thus, recent data demonstrated that maternal obesity was associated with epigenetic changes in both placental leptin and adiponectin systems [32]. More precisely, human maternal obesity was associated with (i) hypermethylation of the DNA in the leptin promoter, (ii) hypomethylation of the DNA in the adiponectin promoter, and (iii) significantly low mRNA and protein expression levels of both leptin and adiponectin receptors in the third-trimester placenta [32]. These results suggest that maternal obesity abrogates the beneficial effects of these two adipokines on placental growth.

Secondly, with regard to the morphological differentiation of CTs, MFSD2 mRNA expression and the fusion index (evaluated by e-cadherin immunostaining) during syncytialization were transiently higher after 24 h and 48 h of cell culture in CTs from the placentas of obese women than in CTs from the placentas of non-obese women [30]. However, the mRNA expression of syncytin-2 (the morphological marker of ST) was similar in control and obese placentas [30]. These data highlighted a dissociation between the trophoblasts’ morphological and biochemical differentiation processes. This dissociation has already been described in the literature and reflects the fact that trophoblast fusion and functional differentiation are regulated in different ways. For example, sphingolipids (and specifically short-chain ceramides) can regulate biochemical trophoblast differentiation but not fusion [35], whereas regulators of fusion can have no effects (or even have opposing effects) on biochemical differentiation [36,37]. Moreover, mitochondria have been linked to CT differentiation; various studies have evidenced morphological and functional changes in mitochondria during CT differentiation. It has been observed that the mitochondria in the human ST are morphologically different from those in the CTs: the mitochondria in the ST are smaller, with a condensed matrix and fewer cristae [38,39]. It has also been suggested that anaerobic metabolism is the main source of ATP production during trophoblast differentiation [40,41]. This could be due to a lower level of mitochondrial ATP production in the ST than in CTs. Moreover, it has been shown that mitochondrial reactive oxygen species (ROS) are the second messengers involved in cell differentiation [42]. Lastly, the morphological and functional changes reported in mitochondria during CT differentiation appear to be associated (at least in part) with elevated steroid production in the ST, since placental cells are the main participants in progesterone synthesis during pregnancy [39,40,43,44]. Recent studies have demonstrated that maternal obesity is associated with a lower mitochondrial content and the disrupted expression of a key regulators of mitochondrial biogenesis and activity, such as transcription factors such as estrogen-related receptor-gamma (ERR-γ), peroxisome proliferator-activated receptors (PPAR)-γ, and PPAR-γ co-activator 1-alpha (PGC-1α) during CT differentiation [30,43,44,45,46]. Taken as a whole, these results highlight the structural, cellular, and molecular mechanisms involved in the placenta’s adaptation to an adverse intrauterine environment.

## 4. Impact of Maternal Obesity on Placental Morphological Characteristics

Some literature data show that human maternal obesity can also affect the morphological characteristics of the third-trimester placenta. An early histological study showed that there were no significant abnormal macroscopic or microscopic placental differences (in terms of placental maturity, the degree of terminal villi formation, and the CD68 and CD14 macrophage counts) between obese and non-obese pregnant women [47]. However, maternal obesity appeared to be associate with greater muscularity in the placental vessel walls [47]. Two subsequent histological studies revealed a number of sex-specific differences: female placentas from obese women were more susceptible to chronic villitis and thrombosis, while male placentas from obese women had more intense villous edema [19,48]. More recently, recent data used an innovative stereological approach to identify placental differences between obese and non-obese pregnant women, showing: (i) a similar volume fraction and surface density for trophoblasts, fetal vessels, mesenchyme, sprouts/knots, and intervillous chamber components in the two groups; (ii) a lower vessel density of the villous tree (as reflected by less intense CD34 and CD31 immunostaining) in the obese group; (iii) a greater frequency of focal subchorial thromboses in the obese group; and (iv) a higher frequency of subchorial fibrin deposits in the obese group [10]. These results suggest that the placental vascular pattern is altered by maternal obesity in the absence of GD. Furthermore, the fibrin deposits specifically observed in placentas from obese women might interfere with perfusion and gas/nutrient exchanges in the intervillous space which, in turn, might result in the chronic placental insufficiency described by Andres et al. [49].

Placental efficiency (defined by the birth weight:placenta weight ratio) might also be modulated by the maternal environment. Indeed, the results of two studies demonstrated that placental efficiency was significantly lower in obese women than in lean women. This observation was particularly true for women with a female fetus (since females presented a greater placental weight than males), suggesting the presence of sex-specific effects of maternal obesity [4,10]. Thus, maternal obesity clearly alters placental structure and efficiency, which might lead to placental dysfunction.

## 5. Impact of Maternal Obesity on Placental Metabolism

Placental nutrient transport depends on the placenta’s size (in particular, the surface area available for exchange), nutrient transporter activity/availability, and the utero– and fetal–placental blood flows [50]. Glucose, amino acids, free fatty acids (FFAs), and cholesterol are essential macronutrients for fetal growth, and each nutrient crosses the placenta through specific transporters and engages different metabolic pathways.

### 5.1. Impact of Maternal Obesity on Placental Glucose Metabolism

Glucose is the most important source of energy for both the placenta and the fetus. Indeed, the fetus is entirely dependent on glucose transfer from the maternal plasma, which is itself conditioned by placental glucose metabolism and transporter expression. Glucose crosses the placenta by facilitated diffusion through glucose transporters 1 and 3 (GLUT1 and GLUT3), which are the two major transporter isoforms expressed at high levels in the human term placenta [51,52]. The results of two recent studies clearly demonstrated that mRNA and protein expression levels of GLUT1 were lower during maternal obesity [10,53]. Moreover, placental metabolism is highly oxidative and “prefers” oxidative metabolism to glycolysis [54]. There is now some evidence to show that maternal obesity increases aerobic glycolysis, and so compromised mitochondrial homeostasis might contribute to fetal acidosis [55,56].

### 5.2. Impact of Maternal Obesity on Placental Amino Acid Metabolism

Amino acids are crucial for the synthesis of important biomolecules (such as nucleic acids and proteins) in the fetal–placental unit. Amino acids are transported from the maternal circulation into the intervillous space by the ST. The active transport across the placenta involves two main isoforms of the L-type sodium-independent neutral amino acid transporter (LAT1 and LAT2) [57,58] and three main isoforms of the A-type sodium-dependent neutral amino acid transporter (SNAT1, SNAT2, and SNAT4); all five isoforms are strongly expressed in the human term placenta [11,59,60]. Glutamate appears to be the most important amino acid substrate for the fetus once it is metabolized into glutamine by the placenta [61]. The results of two recent studies demonstrated that maternal obesity was associated with lower mRNA expression levels of LAT1-2, and SNAT1-2-4 in the placenta [10,11], while a third study did not show any differences in the placental expression of SNAT1 and 4 and even found higher SNAT2 expression levels in placentas from obese women than in placentas from non-obese women [12]. These discrepancies could be explained by a failure to consider the fetal sex or by the inclusion of obese women with a very high BMI (>40 kg/m^2^). However, concerning placental amino acid uptake, a study revealed that maternal obesity was accompanied by a reduced placental SNAT activity [31]. The low levels of placental amino acid transport might also be related (at least in part) to maternal hyperleptinemia. Many studies have shown that leptin favors placental amino acid delivery to the fetus [62]. In addition, it was known that the protein expression level of the placental leptin receptor was significantly lower in obese women than in non-obese women [32]. Hence, one can hypothesize that adaptative leptin resistance arises in the placenta in obese women as a response to maternal hyperleptinemia. Lastly, insulin (a key regulator of placental amino acid transport) might also play a role [63].

Moreover, Ditchfield et al. demonstrated that the placental activity of the taurine transporter TauT was significantly lower in obese women than in normal-weight women [64]. Since the maternal plasma taurine concentration at term was significantly higher in obese women than in non-obese women, the researchers suggested that low placental TauT activity in obesity might be an adaptive, protective response to elevated maternal plasma taurine concentrations. Lastly, using a general approach (metabolomics), a recent study confirmed that maternal obesity altered placental amino acid levels, with higher levels of serine and leucine and lower levels of taurine and lysine [65].

### 5.3. Impact of Maternal Obesity on Placental Lipid Metabolism

#### 5.3.1. Fatty Acids (FAs)

Maternal circulating triglycerides (TGs) are first broken down into FFAs by placental lipases (i.e., lipoprotein lipase and endothelial lipase). The FFAs are then available for uptake into the placenta via FA transport proteins (FATP1 and FATP3) and FA binding proteins (FABPs, cytoplasmic proteins that handle unsaturated FAs and mediate FA metabolism) [66]. The placental lipid content depends on the maternal supply [67]. Therefore, one can expect maternal dyslipidemia in obese women to alter the lipid composition of the placenta itself. Surprisingly, various studies have not revealed any differences in placental total lipid content or lipoprotein lipase mRNA expression between obese and non-obese women [15,68,69]. Nevertheless, it has been reported that maternal obesity was associated with (i) low placental FATP1 mRNA expression and a low proportion of saturated FAs in the placenta from obese women, (ii) low placental FABP1 mRNA expression and an elevated placental content of polyunsaturated free FAs, and (iii) elevated FA translocase FAT/CD36 mRNA expression and, thus, a higher placental content of long-chain polyunsaturated FAs, which have essential structural and functional roles in fetal development [15,68]. These observations suggest that maternal obesity leads to the mobilization and use of specific FAs [68]. Since FATP expression is known to be regulated in a positive feedback loop by FAs and their derivatives, lower expression of FATP1 might therefore be a protective placental adaptation mechanism for limiting excessive nutrient transfer to the fetus. In addition to binding FAs, FABP1 binds a range of hydrophobic molecules such as the PPAR transcription factors. There is evidence to show that PPARs have crucial roles in placental lipid handling and FA metabolism [70,71,72]. However, data in the literature on the relationship between placental PPAR expression and maternal obesity are contradictory. Dubé et al. observed similar PPAR contents in placentas from obese women vs. lean women [15]. In contrast, Calbabuig-Navarro et al. showed that, in placentas from obese women, PPARγ mRNA expression was higher (due to FA synthesis) and PPARα mRNA expression was lower (due to FA oxidation) [73]. Furthermore, two recent general studies evidenced marked, fetal-sex-specific responses in placental FA oxidation, esterification, and transfer capacity to maternal obesity [13,18]. More precisely, it was found that maternal obesity causes (i) lower placental transfer of docosahexaenoic acid (which is critical for fetal growth and brain development) to male fetuses only, (ii) lower placental availability of substrates for β-oxidation (particularly free carnitine, which facilitates the transport of long-chain FAs across the inner mitochondrial membrane) in female placentas only, and (iii) greater enzymic FA esterification activity (i.e., by diacylglycerol-o-acyltransferase 2) in female placentas [13,18]. Furthermore, male (but not female) cultured primary human trophoblast cells isolated from placentas of obese mothers had a greater preference for FA and glucose substrates at baseline. It has also been demonstrated that these substrate preferences were accompanied by a lower placental ability to switch between glucose, FA, and glutamine when oxidation demands increased [74]. Moreover, Mele et al. demonstrated that metabolic flexibility (defined as the cell’s ability to adapt its metabolism to substrate availabilities and energy needs) was lower in the ST from obese women [37]. Lastly, Fattuoni et al.’s metabolomic analysis revealed high levels of palmitic acid and low levels of arachidonic acid and stearic acid in samples from obese women, relative to samples from non-obese women [65].

#### 5.3.2. Sphingolipids

Sphingolipids constitute a large family of lipids and are the main components of biological membranes. They have also been described as bioactive lipids, due to their role as second messengers within the cell [75,76,77]. Ceramides are the predominant precursors of sphingolipids (such as sphingomyelins or gangliosides) and have a crucial role in cell signaling. At present, little is known about the putative roles of these sphingolipids in pregnancy in general and in the human placenta in particular. Some studies have shown that sphingolipids are involved in key cellular processes, such as apoptosis, differentiation, migration, and invasion of trophoblastic cells [35]. These findings suggest that sphingolipids are of importance in placental development. However, to the best of our knowledge, the effects of maternal obesity on the placental sphingolipid profile have not been investigated.

## 6. Impact of Maternal Obesity on Placental Inflammatory/Immune Status

The levels of most of the inflammatory cytokines in the maternal circulation increase significantly during normal pregnancy; this is due, in part, to the secretion of cytokines by the placenta itself [78]. Many researchers have reported that maternal obesity (characterized by chronic, low-grade inflammation) further increases circulating levels of proinflammatory cytokines such as interleukins-1 and -6, tumor necrosis factor alpha (TNF-α), monocyte chemoattractant protein 1, C-reactive protein, and leptin [79,80]. It has even been reported that placental TNF-α levels are abnormally high in female placentas (but not in male placentas) from obese women; again this suggests the presence of fetal sex differences in the placenta’s inflammatory response to obesity [81]. These observations support the hypothesis whereby the mild proinflammatory state associated with normal pregnancy is exacerbated by maternal obesity. However, data in the literature are inconsistent; surprisingly, a number of reports failed to find significantly elevated maternal circulating levels of cytokines in obese pregnant women [82], and one study even observed a significantly low level of placental interleukin-6 secretion and significantly low macrophage/leukocyte infiltration, specifically in GD-free obese women [10]. Furthermore, Nogues et al. reported that maternal obesity did not influence the placenta activation of various major signaling pathways downstream of proinflammatory cytokines, including the Jun kinase, Mitogen-activated protein kinase, and Janus-activated kinase pathways [10].

Taken as a whole, these results suggest that placental inflammatory status could be reduced (at least moderately) by maternal obesity—even in the absence of GD. Since placental inflammatory status is critical for the maintenance of pregnancy, these placental changes might constitute a protective mechanism against the maternal hyperinflammatory environment. There are many possible reasons for these discrepancies, including fetal sex. Thus, the data suggest that heightened inflammation is not a general phenomenon in pregnancies complicated by obesity, and so one cannot rule out the occurrence of this phenomenon in specific subgroups of obese women only. Additional research is needed to better precise these placental adaptations.

## 7. Impact of Maternal Obesity on Placental Oxidative Status

Pregnancy per se is characterized by maternal chronic inflammation, elevated metabolic demand, and thus greater oxidative stress. Mitochondria generate ROS such as the superoxide anion (O_2_^−^) and hydrogen peroxide (H_2_O_2_). Physiological ROS production is essential for some biological processes, such as cell differentiation and the inflammatory response (as described above). However, ROS overproduction and low antioxidant capacity lead to oxidative stress, damage to mitochondria, and the disruption of cell homeostasis [45]. Furthermore, the combination of nitric oxide (NO) and O_2_^−^ produces reactive nitrogen species such as peroxynitrites (ONOO^−^). This powerful oxidant exerts various harmful effects by nitrating transporters, enzymes, and signal transduction molecules [83].

Recent studies have demonstrated that maternal obesity is associated with greater placental oxidative stress. Indeed, levels of markers of oxidative stress (such as lipid peroxidation, protein nitrosylation, and protein carbonylation) are higher in placenta samples from obese women [14,84,85,86]. Moreover, maternal obesity is also associated with damage to mitochondria in placental tissue, with deregulation of mitochondrial DNA content, lower mitochondrial respiration, and less ATP production [16,37,43,44,46,87]. However, the data on the expression and activity of antioxidant enzymes are still subject to debate. Some researchers have reported an elevation of placental antioxidant enzyme levels in maternal obesity [13] while others have not found any differences or have even found abnormally low levels of placental antioxidant enzymes [14,88]. These discrepancies might be due to sexual dimorphism in enzymatic antioxidant defenses. Indeed, male placentas in obese women showed a higher level of oxidative stress and a greater reduction in the activities of both superoxide dismutase and catalase, relative to female placentas [17].

## 8. Impact of Maternal Obesity on the Placental Transcriptome

Overall, gene expression profiling experiments have demonstrated that maternal obesity creates a unique in utero environment—one that impairs the placental transcriptome [89]. Indeed, the results of recent studies have demonstrated a clear difference in the placental transcriptome between obese women and normal-weight women [84,89,90]. More precisely, the placental transcriptome in obese women was characterized by an overall repression of most of the differentially expressed genes. The placental dysregulation observed specifically in samples from the obese women involved genes mainly related to inflammation, immune responses, and lipid metabolism. Interestingly, it was shown that supplementation with unsaturated FAs during pregnancy modifies the placental transcriptome in a sexually dimorphic manner, with female placentas being more responsive [91]; this might reflect greater plasticity in female placentas. In contrast, it has been shown that male placentas express lower levels of the X-linked gene *OGT* coding for O-GlcNAc transferase, which is required in some placental epigenetic processes. More specifically, reduced *OGT* expression could result in male placentas having less of the histone repressive mark H3K2me3 and, thus, being more vulnerable to modifications of the maternal environmental [92,93]. Hence, one can hypothesize that maternal obesity affects placental transcriptome in a sex-specific manner, although data in the literature are lacking.

## 9. Conclusions

Most data in the literature aim to elucidate the influence of maternal obesity on placental functions without taking into account fetal sex. Our review clearly demonstrates the critical importance of fetal sex when evaluating placental responses to metabolic diseases in general and maternal obesity (in the absence of GD) in particular. Indeed, some studies show that the placenta is a transiently plastic organ that adapts its development and metabolism, in a sex-specific manner, in response to an obesogenic environment (Figure 1) (Table 1). These sex-specific placental changes might explain (at least in part) the fetal-sex-dependent outcomes observed in infants born to obese women. However, data concerning certain placental sex-specific adaptations (such as endocrine function and nutrient exchange) are currently missing. Therefore, further investigations are needed to completely understand the involvement of fetal sex in placental adaptive changes and to support the notion of sexual dimorphism in response to maternal obesity.

Maternal obesity exposes the placenta to a lipotoxic environment that might alter placental functions and the offspring’s health via changes in placental nutrient transporter expression, mitochondrial function, lipid metabolism, and oxidative stress levels. Moreover, sex-specific placental responses have been described. Female placentas from obese women are characterized by lower levels of FA oxidation, higher levels of FA esterification, and higher levels of inflammation, and appear to adapt more easily to a lipotoxic maternal environment. Nevertheless, it has been observed that female fetuses have larger amounts of adipose tissue. Male placentas are characterized by lower expression of antioxidant enzymes, a moderate level of inflammation, and lower docosahexaenoic acid transfer. The latter might be related to the poor neurodevelopmental outcomes observed specifically in male fetuses. The metabolic alterations caused by maternal obesity might contribute to poor placental endocrine function and vascular alterations. Lastly, these various placental modifications might explain (at least in part) the long-term risks of metabolic and cardiovascular diseases observed in the offspring of obese women.

GD: gestational diabetes; GLUT: glucose transporter; SNAT: A-type sodium-dependent neutral amino acid transporter; LAT: L-amino acid transporter; FATP: fatty acid transport protein FABP: fatty acid binding protein; FAT/CD36: fatty acid translocase; FA: fatty acid; SFA: saturated fatty acid; DHA: docosahexaenoic acid; TNF-α: tumor necrosis factor alpha.

## Figures and Tables

**Figure 1 ijms-24-09770-f001:**
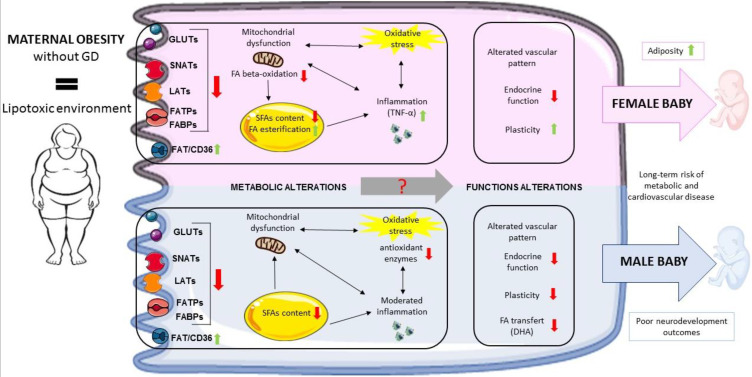
Summary of the impact of maternal obesity on placental adaptive changes, according to the fetal sex.

**Table 1 ijms-24-09770-t001:** Summary of the impact of maternal obesity on placental functions at the molecular level, according to the fetal sex.

Placental Function	Maternal Obesity Impact
Endocrine function	hCG Leptin ProgesteroneLeptin receptorAdiponectin receptor	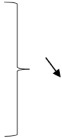
Mitochondrial function	ERR-γ expressionPPAR-γ/α expressionPGC-1α expressionMitochondrial DNA contentMitochondrial respiration ATP production	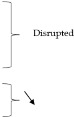
Oxidative stress	Lipid peroxydationProtein carbonylationProtein nitrosylationCatalaseSuperoxide dismutase	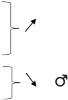
Inflammation	Jun kinase Mitogen-activated protein kinase Janus-activated kinase TNF-α	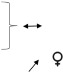
Epigenetic changes	Leptin DNA methylation Adiponectin DNA methylationH3K2me3	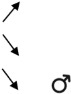
FA esterification	Free carnitineDiacylglycerol-o-transferase	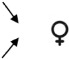

hCG: human chorionic gonadotropin; ERR-γ: estrogen-related receptor-gamma; PPAR: peroxisome proliferator-activated receptors; PGC-1α: PPAR-γ co-activator-1 alpha; TNF-α: tumor necrosis factor alpha; FA: fatty acid.

## Data Availability

No new data were created or analyzed in this study. Data sharing is not applicable to this article.

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
