# Peer review of "Human Placental Adaptive Changes in Response to Maternal Obesity: Sex Specificities"

_ijms, 2023, doi:10.3390/ijms24119770_

Round 1
Reviewer 1 Report
This is a very nice review regarding the placental adaptive changes in response to maternal obesity.
However, the title and abstract suggest that there will be a distinct focus on sex specific differences. While the figures at the end do have information about sex specific differences, it feels as though the few sex specific differences discussed in the text get overwhelmed by the significant amount of data regarding the general mechanisms of placental adaptation in response to maternal obesity. Obviously there is not much that has been publised on sex specific differences (which I appreciate) I just worry that this novel aspect of the review is getting lost. There has recently been quite a focus in large animal models in sex differences in placental development, structure and function - perhaps you could maybe bring in some data from large animals or rodent models of obesity?
Few minor things but no major issues
Reviewer 2 Report
The current review summarized the impact of maternal obesity on various characteristics of human placenta. The authors also mentioned the sex-specific response to maternal obesity. The overall presentation is very good, and the cited references are highly relevant. However, the main concern is that there is lack of confidence and evidence to support the notion of sexual dimorphism in response to maternal obesity. In particular, there is lack of evidence regarding sex-specific responses reported on ‘endocrine function’ and ‘glucose transport/metabolism’ an ‘amino acid transport’ sections while there are very few studies reported on other sections. So based on this, one could be reluctant to assume that fetal sex would be playing a critical role demonstrated in the conclusion and title of this manuscript.
Please add the line number in the manuscript.
Please see specific comments below:
1.Introduction
1st Para, Line 3 – need references
2nd Para, line 5 – what is the offspring age? It would be worth to add the information
2nd Para, last sentence, is the sexually dimorphism in neurodevelopmental disorders due to maternal obesity? please clarify.
3rd Para, suggesting merge this short paragraph into the next one.
3rd Para, first and the last sentences both need references
2.Placental development:
1st Para: it would be helpful to provide information regarding the type of human placenta with references.
3.Impact of maternal obesity on placental development: trophoblast differentiation
Suggesting changing this title to “Impact of maternal obesity on endocrine function” – the authors mentioned this in the Abstract and should follow that flow. Also, the first paragraph needs to be shortened.
How about the sex-specific change in response to maternal obesity? The discussion is missing for the endocrine function section.
1st Para, the first, third, and the second last sentences – both need references.
“Thus, Nogues et al. demonstrated that…” seems incorrect reference style, please check here and throughout the manuscript.
1st Para, second last sentence – mRNA or protein expression? Please clarify.
2nd Para, first sentence – reference needed.
4th Para, suggest combining this short para into the previous one.
4. Impact of maternal obesity on placental structure and efficiency
Suggest changing the subtitle to Impact of maternal obesity on placental morphological characters so it aligns with your previous description in the Abstract.
1st Para, second sentence – need a reference.
2nd Para, 2nd sentence – Interesting data. Is the lower placental efficiency in obese women due to lower birth weight or greater placental weight? Suggesting adding that information and clarify. The last sentence - this might be due to lower birth weights of females than males, regardless of maternal environment, as females are generally born lighter than males.
3rd Para – again, please eighter remove the short para or merge it into the previous one. Please check it throughout the manuscript.
5.1. Impact of maternal obesity on placental glucose metabolism
It would be interesting to know if there are any literature reporting changes in glucose uptake/ transport activity due to maternal obesity.
There is a lack of references cited regarding the sex-specific change in glucose metabolism due to maternal obesity. Is this because of the lack of studies?
5.2. Impact of maternal obesity on placental amino acid metabolism
1st Para – Could higher SNAT2 expression levels in placentas from obese women be associated with placental adaptative strategies to enhance the nutrient transport in obese women? Please elucidate this possibility.
Again, is there any evidence demonstrating the change in utero-placental amino acid uptake activity/levels? It would be helpful to include this.
Also, there is a lack of cited references showing the sex-specific change in amino acid metabolism due to maternal obesity. Similar to the glucose section, and as this is one of the review focuses, it would be helpful to add those studies if exists.
5.3.1. Fatty acids (FAs)
“More precisely, it was found that mater-nal obesity causes (i) lower placental transfer…” and the following sentence – need references.
6. Impact of maternal obesity on placental inflammatory/immune status
1st Para, 1st sentence – is this for normal pregnancy or obesity conditions? Please clarify.
8. Impact of maternal obesity on the placental transcriptome
3rd sentence – is there any evidence on DNA methylation changes as most of the genes were downregulated by maternal obesity? Also, is the repression of expression of genes associated with inflammation and oxidative stress? This should be upregulated as you have mentioned in previous sections.
Conclusion
The conclusion section would benefit from rewording. I disagree that, based on the evidence from this review, fetal sex would be playing a critical role as there are very few past literatures supporting this hypothesis.
The focus of this review mainly lies on the impact of maternal obesity on various placental characteristics, and some of those changes of responses could be supported by fetal sex as demonstrated. However, there is still lack of evidence to support the sexual dimorphism in responses to maternal obesity. The authors could highlight and hypothesize the sex effect as it warrants further investigation.
English language is very good
Round 2
Reviewer 2 Report
I thank the authors for the detailed response and revision of the manuscript. Congratulations on the great work.